# Challenges Pertaining to the Optimization of Therapy and the Management of Asthma—Results from the 2023 EU-LAMA Survey

**DOI:** 10.3390/biomedicines13081877

**Published:** 2025-08-01

**Authors:** Michał Panek, Robab Breyer-Kohansal, Paschalis Steiropoulos, Peter Kopač, Monika Knopczyk, Tomasz Dębowski, Christer Janson, Maciej Kupczyk

**Affiliations:** 1Department of Internal Medicine, Asthma and Allergy, Medical University, 90-153 Łódź, Poland; maciej.kupczyk@umed.lodz.pl; 2Department of Respiratory and Pulmonary Diseases, Clinic Hietzing, Vienna Healthcare Group, 1130 Vienna, Austria; robab.breyer-kohansal@lunghealth.lbg.ac.at; 3Department of Respiratory Medicine, Medical School, University General Hospital, Democritus University of Thrace, 68100 Alexandroupolis, Greece; steiropoulos@yahoo.com; 4Department of Allergology and Clinical Immunology, University Clinic of Pulmonary and Allergic Diseases, 4204 Golnik, Slovenia; peter.kopac@klinika-golnik.si; 5Medical Department, Chiesi Poland Sp. z o.o., 02-305 Warsaw, Poland; m.knopczyk@chiesi.com (M.K.); t.debowski@chiesi.com (T.D.); 6Department of Medical Sciences, Respiratory, Allergy and Sleep Research, Uppsala University, 751 85 Uppsala, Sweden; christer.janson@medsci.uu.se

**Keywords:** asthma management, adult, severe asthma, GINA, SITT, LAMAs, TRIPLE therapy

## Abstract

**Background:** Treatment compliant with the Global Initiative for Asthma (GINA) can promote more effective disease control. Single-inhaler triple therapy (SITT) is one method that is used to optimize therapy in this context, but TRIPLE therapy is still employed by physicians to a limited extent. **Objective:** This study aimed to describe the factors influencing challenges in optimizing asthma therapy. **Methods:** A 19-question survey, created via the CATI system, was distributed among pulmonologists, allergologists, general practitioners, and internal medicine specialists in Poland, Greece, Sweden, Slovenia, and Austria. **Results:** Statistically significant percentage differences in the use of TRIPLE therapy in the context of asthma management were observed among countries as well as between pulmonologists, allergists, and other specialists. Overuse of oral corticosteroids (OCSs) to treat nonsevere and severe asthma in the absence of an approach that focuses on optimizing inhalation therapy among asthma patients receiving TRIPLE therapy was observed in different countries as well as among physicians with different specialties. Twenty elements associated with the challenges involved in diagnosing and managing difficult-to-treat and severe asthma were identified. Six clinical categories for the optimization of asthma therapy via SITT were highlighted. The degree of therapeutic underestimation observed among severe asthma patients was assessed by comparing actual treatment with the recommendations of the GINA 2023 guidelines. **Conclusions:** Physicians of various specialties in Europe are subject to therapeutic inertia in terms of their compliance with the GINA 2023 guidelines.

## 1. Introduction

Recently, interest in the use of triple combination therapy involving inhaled corticosteroids (ICSs), long-acting β2-adrenoceptor agonists (LABAs), and long-acting muscarinic antagonists (LAMAs) among patients with chronic obstructive respiratory diseases (CRDs) has increased significantly [1,2]. In particular, the 2023 Global Initiative for Asthma (GINA) recommended the combination of an ICS with a LABA and a LAMA to treat severe forms (i.e., stage 5) of asthma. Multiple pivotal clinical trials have revealed that administering fixed-dose ICS/LABA/LAMA combinations can provide patients with improved disease control and reduce the frequency of asthma exacerbations in comparison with the use of single and dual combination therapy, namely, LAMAs administered as a monocomponent and ICS/LABA or LABA/LAMA combinations [3,4,5,6]. Triple therapy is effective among patients who have experienced one (or more) asthma exacerbations during the previous year, have been treated with at least a medium dose of ICS-LABA, and have experienced an exacerbation despite treatment [7].

Furthermore, ICSs, LABAs, and LAMAs have synergistic anti-inflammatory and bronchorelaxant effects; notably, the synergistic interactions induced by a triple combination have been investigated, as not all asthma patients can benefit from the effects of single-inhaler triple therapy (SITT) [8,9]. Asthma, especially severe asthma, is a considerable challenge for both patients and healthcare professionals [10]. This type of management, which is consistent with the recommendations of the GINA, improves control, reduces the number of exacerbations, improves individuals’ quality of life, and minimizes asthma complications. Nevertheless, among patients with severe asthma, this approach may not only improve their clinical and functional outcomes but also delay the prescription of OCSs and biologics, thereby positively influencing healthcare costs [1,10,11,12].

The EU-LAMA survey aimed to evaluate the perspectives of real-life specialists on implementing ICSs/LABAs/LAMAs in fixed-dose combinations as part of a strategy to optimize therapy and manage asthma in clinical practice.

## 2. Materials and Methods

The Bioethical Committee at the Medical University of Lodz, Poland, approved this study on 18 April 2023 (scientific project No. RNN/88/23/KE).

The medical research survey EU-LAMA was conducted in Poland, Greece, Sweden, Slovenia, and Austria by a contract research organization (CRO) company (Biostat Sp. z o.o.) via its proprietary electronic software—the CATI system; http://www.cati-system.pl/ (accessed on 5 April 2023)—in the form of an electronic computer-assisted web interviewing (CAWI) survey, which was designed by Chiesi Poland in collaboration with the CRO. Before beginning the study, the CRO calculated the expected sample size based on the number of medical professionals with specialties of interest in each country. Subsequently, the questions included in the survey were translated into local languages, and the project design, in line with local regulations, was submitted to ethics committees. Physicians were recruited by the CRO via a structured panel of verified, licensed professionals in each country. Recruitment was conducted through targeted email invitations sent to physicians identified from national registries and medical panels, ensuring representativeness by specialty and geography. Participation was voluntary and based on an informed agreement to complete the full CAWI questionnaire.

The EU-LAMA survey, consisting of a total of 19 questions, aimed to assess physicians’ perceptions of the current role played by LAMAs in asthma treatment. The initial three questions pertained to participants’ specialization and primary workplace as well as the size of the city in which their practices were located. The subsequent 16 questions focused on participants’ opinions regarding the research questions.

In this study, the use of triple therapy was defined as physicians’ self-reported prescription of a fixed-dose combination of ICSs/LABAs/LAMAs for asthma treatment. Responses were based solely on physicians’ clinical practices and therapeutic decisions and did not consider patient-reported data or administrative information on prescription fulfillment.

Statistical details concerning the physicians who participated in the EU-LAMA survey can be found in the online repository.

Out of 767 eligible physicians contacted across the five participating countries, 630 agreed to participate in the EU-LAMA survey and completed the full questionnaire. The sample size was estimated by the research company Biostat Sp. z o.o. based on stratified data from national physician registries. The margin of error was calculated at a 95% confidence level.

The frequency distribution of categorical data was summarized by determining the counts for each category alongside the corresponding percentages. These results are presented in tabular format, thus ensuring clarity and facilitating a straightforward interpretation of the data.

The relationships observed among categorical variables were analyzed via the chi-square test or Fisher’s exact test when assumptions required for the chi-square test were not met. Post hoc analyses were performed when statistically significant differences were identified. For unpaired data, pairwise comparisons with Bonferroni correction for multiple testing were used to identify specific group differences. McNemar’s test was used to investigate the paired data associated with Issue 6 and also to conduct subgroup analyses (based on participants’ specializations and countries), and Bonferroni correction was used to adjust for multiple comparisons.

The level of significance was set to *p* = 0.05, but the results were statistically significant at the *p* = 0.01 and *p* = 0.001 levels. In the figures, *p* values that indicate statistically significant results are highlighted in bold, while cases of *p* < 0.001 are always indicated by the notation *p* < 0.001.

The analysis focused on assessing therapeutic undertreatment among patients who should be managed at GINA Step 5 (see the Online Repository in Appendix A).

All calculations were performed with the assistance of the R statistical package, version 4.4.1, which was also used for creating graphs.

## 3. Results

The EU-LAMA survey involved 630 physicians with various specialties (mainly pneumonologists, who accounted for 58.7% of the total, and allergologists, who accounted for 15.7%) who were recruited from Poland (57.6%), Greece (27%), Sweden (6.3%), Slovenia (5.4%), and Austria (3.7%). A total of 91.6% (*N* = 577) of these physicians reported using TRIPLE therapy in the context of asthma management. The percentages of physicians who used TRIPLE therapy for asthma management differed significantly by country and specialization. Such variations in asthma management methods were not observed for participants’ primary workplace. Detailed descriptive data are presented in Appendix A. OCSs are still used by physicians for patients with nonsevere and severe asthma before the optimization of ICSs/LABAs/LAMAs in the form of fixed-dose combination therapy, with the highest percentages in this context were observed among internal medicine physicians, at 4.5% and 10.6%, respectively. This procedure was observed most frequently in Sweden, where it was used among 25.0% of nonsevere asthma patients and 10.0% of severe asthma patients. A comparison of the use of OCSs versus TRIPLE therapy—taking into account the specialization and country of the physician—is presented in Figure 1 and Figure 2.

According to physicians, the main options to optimize uncontrolled asthma therapy for patients receiving medium or high doses of ICSs/LABAs are to add LAMAs to current treatments (89.7%) and to increase the dose of ICSs to the recommended maximum (84.9%). The methods used to optimize treatment for patients with uncontrolled severe asthma who received medium or high doses of ICSs and LABAs—taking into account the specialization and country of the physician—are presented in Figure 3 and Figure 4.

The most important challenges encountered in the context of asthma therapy and physician–patient collaboration in the management of difficult-to-treat asthma and severe asthma are presented in Appendix A), while factors that can enhance physician–patient collaboration in the context of managing severe and difficult-to-treat asthma were identified and presented in Table 1.

No significant differences for the variables presented above were observed with regard to physicians’ specializations or countries of origin.

The patient’s path toward optimized inhalation therapy are described, particularly with respect to factors that may significantly contribute to the proper management of asthma, for which detailed descriptive data are presented in Appendix A.

The most recent analysis focused on assessing therapeutic undertreatment among patients who should have been managed at GINA Step 5. This issue is often associated with key clinical decisions, such as whether to add a LAMA to current treatments, increase the dose of ICSs to the recommended maximum, initiate oral corticosteroids (OCSs), or refer patients for biological treatment. Despite the availability of these options, physicians frequently continue to treat patients at GINA Step 4, even when their condition clearly necessitates escalation to Step 5.

The analysis was based on four specific responses to survey questions 6 and 7 and involved reassigning positive responses pertaining to GINA Step 4 to their corresponding categories in GINA Step 5. This approach facilitated the creation of two distinct datasets: one that reflects real-world treatment practices and one that represents guideline-based treatment.

Table 2 highlights the discrepancy between real-world therapeutic practices and the guidelines outlined in GINA 2023, which is especially evident with respect to increasing the dose of ICSs to the recommended maximum, where only 28.4% of respondents reported using this method, in comparison with the 84.9% rate recommended by the relevant guidelines. A smaller yet substantial gap was also observed with regard to the addition of LAMA to current treatments, which was observed in 79.4% of real-world settings (as opposed to the recommended rate of 89.7%). Furthermore, McNemar’s test revealed statistically significant differences among all the variables included in the analysis, thus emphasizing the divergence between actual practices and the recommendations provided by the relevant guidelines.

## 4. Discussion

At present, several options, which are described in the guidelines for specific steps, are available to patients who cannot control asthma through the combination of ICSs and LABAs [11,12,13]. Although the post-TRIPLE therapy position has been well documented in clinical trials and the GINA 2023 recommendations, as indicated by the EU-LAMA survey, not all patients can continue to benefit from this therapeutic option (Figure 1). Large differences were observed in patients’ access to SITT across different countries, resulting from differences in medicine reimbursement practices. Interestingly, these differences were also observed among specialists who optimized therapy via the TRIPLE combination. Figure 3 presents the differences in the approaches used by different specialists to optimize asthma treatment. Although all of these approaches were based on one common GINA guideline, they were treated differently and were not aligned with the recommendations. No influence on the use of TRIPLE therapy in the context of asthma management by primary workplace was observed (see Appendix A). It is important to note that the survey design and data interpretation in this study were primarily based on the GINA 2023 report, which served as the core clinical framework during the research conducted in 2023. The reference to the GINA 2024 update was included to emphasize the continuity and consistency of recommendations, particularly regarding the role of TRIPLE therapy in Step 5 asthma management.

The overuse of OCSs among nonsevere asthma patients (an average of 4.9%) and severe asthma patients (an average of 3.5%) was questionable, and significant discrepancies among specialties and countries were observed (further details are presented in Figure 1 and Figure 2). The results indicate large discrepancies in the use of OCSs across individual specialties and countries, which are not justified by the rules of asthma therapy according to the GINA 2023 guidelines. We present the reasons for this phenomenon based on data presented in Appendix A, indicating the ongoing need to add LAMA medicines to the treatment of uncontrolled asthma for patients receiving medium or high doses of ICSs and LABAs, increase the dose of ICSs to the recommended maximum and refer patients for biological treatment (e.g., through monoclonal antibodies, i.e., MABs) regardless of the physician’s specialization, which seems to be in line with the principles of asthma therapy optimization with respect to GINA Step 5. Some differences in the optimization of treatment among patients with uncontrolled severe asthma who received medium or high doses of ICSs and LABAs were observed among individual countries; these discrepancies result from different policies for national public health systems or insurance systems, such as the limitation of health insurance to the reimbursement of costs (Slovenia) or the narrowing of the indications for the administration of MABs in severe asthma cases through the imposition of various restrictions, such as, for example, “only among patients with severe asthma who are dependent on OCS”, “among patients with eosinophils above 300”, or “among patients with high eosinophil levels and high FeNO” (Sweden). The physicians included in this research generally highlighted key challenges in asthma management regardless of their specialization or the country in which they practiced professionally (exceptions are presented in Appendix A).

According to physicians with different specializations, the main problems encountered in the context of efforts to optimize difficult-to-treat and severe asthma lie in patients’ compliance with therapeutic recommendations and errors in inhalation therapy. Challenges concerning the diagnosis and management of difficult-to-treat and severe asthma by pneumonologists, allergologists, internal medicine specialists, and other specialists include the following: a lack of experience with biological treatment; a focus on inhalation therapy; a lack of new therapies; and a lack of experience with the use of triple medicine therapy. These challenges clearly indicate the need for dedicated specialists to provide therapy to patients dealing with severe asthma in specialized centers. These differences were also evident across different countries, and further details can be found in Appendix A. Interestingly, the challenges encountered in the process of diagnosing and treating asthma do not pertain to the same issues that were associated with access to tests, including X-ray examination and complete blood count based on differential and laboratory tests.

There are significant challenges concerning the optimization of asthma therapy via the use of triple therapy by different specialists in patients for whom high doses of ICS and/or oral steroids are not recommended; patients with “severe but more chronic” inflammation, persistent obstruction, lower FEV1, and frequent exacerbations; and patients who do not meet the criteria for biological therapy. Differences in this regard were also observed among different countries in Europe (see Appendix A). The data indicate that physicians with different specialties seek to optimize asthma therapy through the use of triple therapy only when they are convinced that their patient is more seriously ill, i.e., when a high dose of ICSs is insufficient to control asthma symptoms and the patient exhibits severe inflammation or low values of spirometric parameters. However, this approach is unnecessary because GINA proposes that therapy and asthma control should be optimized at each step [1,12].

The most interesting finding of this research, according to the authors, pertains to the statistical analysis (details of which are presented in the Methodology Section) of the degree to which the recommendations of the GINA 2023 guidelines have been implemented in real life. Table 2 highlights the underestimation of LAMA therapy, the underestimation of high-dose ICSs, the underestimation of the use of OCSs, and the ineffectiveness of referring patients with severe asthma for biological treatment (e.g., MABs) in comparison with the recommendations of the GINA guidelines. Exploration of such therapeutic underestimation in severe asthma cases—i.e., a comparison of actual treatment in real-world practice with the recommendations of the GINA 2023 guidelines—reveals that pulmonologists use LAMAs more often than allergologists, but pulmonologists still do so less often than is recommended by the guidelines. In turn, allergologists are more likely to use higher doses of ICSs than pulmonologists, but they still do so less frequently than is recommended by the GINA 2023 guidelines. On the other hand, OCS therapy is used most often by internal medicine specialists, who do so significantly more often than pulmonologists or allergologists. Internal medicine specialists also refer patients for biological treatment (e.g., MABs) least often, and they also do so less often than pulmonologists and allergologists. Significant differences among countries were observed with respect to the addition of LAMAs to current treatments (*p* = 0.011), the addition of oral corticosteroids (OCSs) to current treatments (*p* < 0.001), and the referral of patients for biological treatment (e.g., MABs) (*p* < 0.001). Further details are presented in Appendix A, which are included in the Appendix A. This situation results from differences in national medicine reimbursement systems pertaining to national public health funds and the possibility that individual specialists may be responsible for prescribing specific groups of medicines. For example, in Greece, only pulmonologists and allergologists prescribe biological medicines according to the Mandatory Therapeutic Protocols. In Austria, for biological medicine, certain strictly defined approval criteria must be met for each given medicine, e.g., severe asthma or triple therapy; if these criteria are met, no restrictions for the prescription of such medicines are stipulated. In Slovenia, biological medicines are introduced based on the opinion of a panel of pulmonologists specializing in severe asthma in three hospital centers throughout the country. In Sweden, medicines are distributed throughout the country, and their use is naturally limited because clinics are required to maintain a certain budget. Finally, in Poland, medicines are prescribed to patients according to the indications and contraindications of the medicine program developed by the Ministry of Health.

The differences in asthma management approaches observed across medical specialties (e.g., pulmonologists, allergists, and internists) and between the countries included in the study, as discussed in detail above, stemmed from various factors, including the following:Prescribing patterns (e.g., LAMA addition, ICS dose escalation, or overuse of oral corticosteroids);Access to biologic therapies;Familiarity with guideline-based treatment escalation according to GINA Step 5.

We would like to emphasize that this variability may result from differences in specialty training, national reimbursement policies, and clinical experience.

Also of note is that socioeconomic differences, such as healthcare funding models, drug reimbursement policies, and patients’ out-of-pocket costs, may influence both physicians’ prescribing decisions and patients’ adherence to treatment. Although our study focused on physicians’ perspectives, it is important to note that treatment outcomes are strongly shaped by the broader healthcare system context and individual patient circumstances. We recommend that future research includes complementary qualitative and patient-centered studies in this area.

In addition, our findings suggest that suboptimal adherence to GINA guidelines in clinical practice may be multifactorial. Potential contributors include therapeutic inertia, administrative restrictions on therapy escalation (e.g., limited access to biologics), time constraints during consultations, and insufficient availability of diagnostic tools such as spirometry or biomarker testing. Some physicians may also lack confidence in newer treatment options due to limited training or experience. These factors, combined with systemic and socioeconomic barriers, explain the observed gaps between guideline recommendations and real-world practice.

The limitations of our work include the fact that we collected data only from physicians who agreed to answer the questions asked as part of the interviews we conducted, and only physicians from certain European countries participated in this study. The CAWI method also entails certain disadvantages in terms of limited representativeness (i.e., this method is less accessible to older people or those with limited internet access) and a lack of control (i.e., the respondents may not have understood the questions or answered them truthfully).

## 5. Conclusions

Differences were observed in terms of patients’ access to TRIPLE therapy in the context of asthma management depending on physicians’ specializations and countries, the overuse of OCSs to treat nonsevere and severe asthma patients, and therapeutic underestimation among severe asthma patients; furthermore, a number of factors that influence physician—patient cooperation were identified.

## Figures and Tables

**Figure 1 biomedicines-13-01877-f001:**
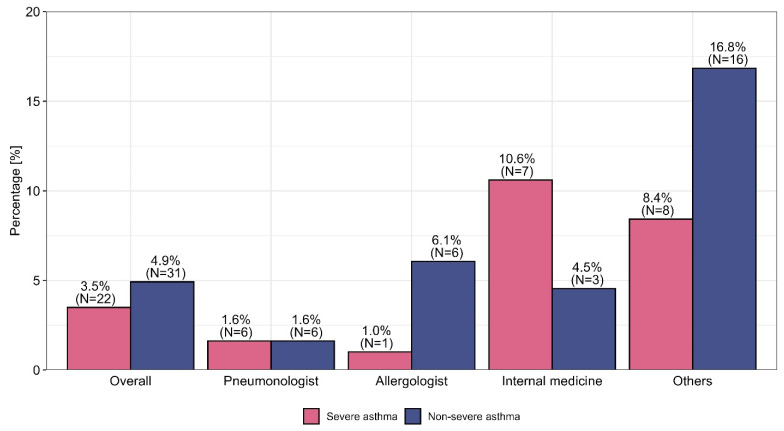
The use of OCSs versus TRIPLE therapy among severe and nonsevere asthma patients by specialization.

**Figure 2 biomedicines-13-01877-f002:**
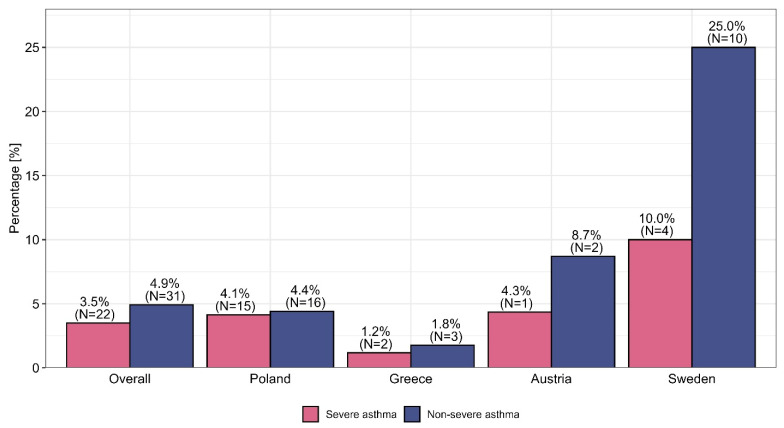
The use of OCSs versus TRIPLE therapy among severe and nonsevere asthma patients by country.

**Figure 3 biomedicines-13-01877-f003:**
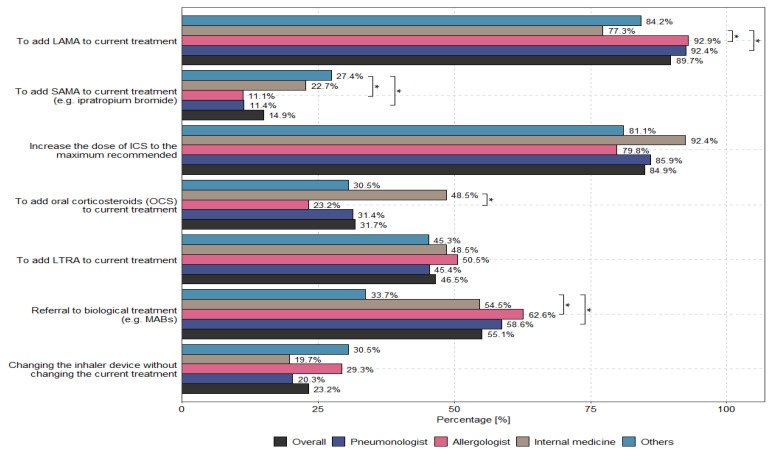
Optimization of treatment by specialization in patients with uncontrolled severe asthma who are on medium- or high-dose ICSs and LABAs. The asterisk (“*”) indicates statistically significant differences between groups (*p* < 0.05) in the figure.

**Figure 4 biomedicines-13-01877-f004:**
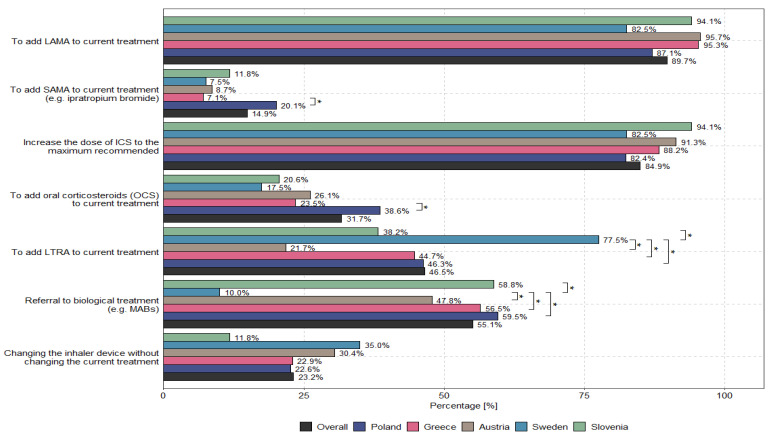
Optimization of treatment by country in patients with uncontrolled severe asthma who are on medium- or high-dose ICSs and LABAs. The asterisk (“*”) indicates statistically significant differences between groups (*p* < 0.05) in the figure.

**Table 1 biomedicines-13-01877-t001:** Factors that can enhance physician–patient collaboration in the context of managing severe and difficult-to-treat asthma.

Variable	Overall (*N* = 630)
Communication between specialists and primary care physicians	58.7% (*N* = 370)
Easy access to biological therapy (lower price)	52.2% (*N* = 329)
Modification of criteria for patient inclusion in biological therapy	36.5% (*N* = 230)
Established position of the TRIPLE drug therapy in relevant guidelines/recommendations	48.1% (*N* = 303)
New inhalation therapies	35.1% (*N* = 221)
Time for patient education/contract for educator or pulmonary nurse	72.9% (*N* = 459)
E-device (device monitoring, e.g., the number of doses inhaled)	46.8% (*N* = 295)
Education of medical personnel	43.7% (*N* = 275)
Good communication with patients	66.0% (*N* = 416)
Psychological support for patients	45.6% (*N* = 287)

**Table 2 biomedicines-13-01877-t002:** Therapeutic underestimation in severe asthma cases: a comparison of actual treatment with the recommendations of the GINA 2023 guidelines.

Variable	Real-World Treatment (*N* = 630)	Guideline-Based Treatment (*N* = 630)	Test	*p* Value
Adding LAMAs to current treatments	79.4% (*N* = 500)	89.7% (*N* = 565)	McNemar	<0.001
Increasing the dose of ICSs to the recommended maximum	28.4% (*N* = 179)	84.9% (*N* = 535)	McNemar	<0.001
Adding oral corticosteroids (OCSs) to current treatments	27.1% (*N* = 171)	31.7% (*N* = 200)	McNemar	<0.001
Referring patients for biological treatment (e.g., MABs)	50.8% (*N* = 320)	55.1% (*N* = 347)	McNemar	<0.001

Detailed descriptive data, by specialization and country, are presented in Appendix A.

## Data Availability

The data has not been published yet.

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
