# Peer review of "Challenges Pertaining to the Optimization of Therapy and the Management of Asthma—Results from the 2023 EU-LAMA Survey"

_biomedicines, 2025, doi:10.3390/biomedicines13081877_

Round 1

Reviewer 1 Report

Comments and Suggestions for Authors

I have the opportunity to read a submission to another journal, and now this manuscript has been improved. The importance of SCS overuse was registered in graphics.

The authors could inform the % of physicians who agreed to answer the questions.

Author Response

We sincerely thank the Reviewer for recognizing the improvements made in the revised version of the manuscript and for the valuable suggestion.

Comment 1: The authors could inform the % of physicians who agreed to answer the questions.

Response 1: In response to the comment regarding the proportion of physicians who agreed to participate in the survey, we have now added this information to the Materials and Methods section. Specifically, we state:

“Out of 767 eligible physicians contacted across the five participating countries, 630 agreed to participate in the EU-LAMA survey and completed the full questionnaire.”

This corresponds to a response rate of approximately 82.1%. This information provides greater transparency regarding the response rate and supports the representativeness of our data.

We also appreciate the Reviewer’s recognition of the enhanced visual presentation of SCS overuse in the figures and thank you for your positive feedback.

Reviewer 2 Report

Comments and Suggestions for Authors

I read this paper with great interest. This manuscript addresses an important issue, the underuse of SITT in asthma management across Europe  and has robust multicountry survey data. The paper is well-written and it's clear.

However, I have some concerns which must be addressed: 

- Ref 1 and 12 cite GINA 2023 and 2024. Please clarify why both reports are cited and integrate their recommendations coherently.

-The methods section should describe how the physician sample was recruited (e.g. email invitations, random selection, professional associations).

-Many important figures are relegated to the supplementary material. Why don't you integrate key figures/tables (e.g. Figure S1, S4, Table 2) into the main manuscript? 

-Clarify how triple therapy use was defined in the survey: was it physician prescription, patient-reported use, or administrative data?

- I found some English errors in the manuscript. Please have a deep language revision. 

  •  
Comments on the Quality of English Language

moderate revision required 

Author Response

We sincerely thank the Reviewer for the positive assessment of our work and for the thoughtful and detailed comments, all of which have contributed to further improving the clarity and completeness of our manuscript. Below, we provide point-by-point responses.

Comment 1: Ref 1 and 12 cite GINA 2023 and 2024. Please clarify why both reports are cited and integrate their recommendations coherently.

Response 1: Thank you for this important observation. We have now clarified in the Discussion section that the GINA 2023 report was used as the primary clinical framework during survey design and implementation, as the study was conducted in 2023. The GINA 2024 update, referenced as Ref. 12, was added to indicate the continuity and consistency of recommendations, especially regarding the role of SITT in Step 5 asthma management. We now integrate both sources more clearly and explicitly state their respective relevance in the revised manuscript.

Comment 2: The methods section should describe how the physician sample was recruited (e.g. email invitations, random selection, professional associations).

Response 2: We appreciate this suggestion. We have updated the Materials and Methods section to include the following clarification:

“Physicians were recruited by the contract research organization (Biostat Sp. z o.o.) via a structured panel of verified, licensed professionals in each country. Recruitment was conducted through targeted email invitations sent to physicians identified from national registries and medical panels, ensuring representativeness by specialty and geography. Participation was voluntary and based on informed agreement to complete the full CAWI questionnaire.”

Comment 3: Many important figures are relegated to the supplementary material. Why don't you integrate key figures/tables (e.g. Figure S1, S4, Table 2) into the main manuscript? 

Response 3: We agree that certain figures and tables from the supplementary files provide essential context and deserve to be featured in the main body of the manuscript. In response, we have now:

  1. Moved Figure S4 and Figure S5 into the Results section of the main text,  
  2. Revised figure legends to ensure clarity and consistency.

These changes improve the flow of the manuscript and allow readers to directly access critical data without referring to supplementary files.

Comment 4: Clarify how triple therapy use was defined in the survey: was it physician prescription, patient-reported use, or administrative data?

Response 4: Thank you for requesting clarification. We have now added the following sentence to the Methods section:

“Triple therapy use was defined in the survey as the physician’s self-reported prescription of a fixed-dose combination of ICS/LABA/LAMA for asthma management. Responses were based exclusively on physicians’ clinical practice and prescribing behavior, not on patient self-reports or administrative prescribing data.”

Comment 5: Language and grammar revision:

Response 5: We appreciate this helpful remark. The entire manuscript has undergone professional English language editing by a native speaker with scientific writing expertise. We focused on simplifying complex sentences, correcting grammatical issues, and ensuring uniform terminology. We believe these improvements significantly enhance the clarity and readability of the manuscript.

Once again, we thank the Reviewer for the constructive and insightful feedback. All suggested revisions have been implemented with care and have meaningfully strengthened our work.

Reviewer 3 Report

Comments and Suggestions for Authors

The manuscript entitled “Challenges Pertaining to the Optimization of Therapy and the Management of Asthma – The Results of the EU-LAMA Survey 2023” presents a detailed analysis of the EU-LAMA survey conducted across five European countries. The article addresses real-world gaps in asthma control, with particular emphasis on the use of single-inhaler triple therapy (SITT). The topic is timely and relevant, especially in light of the ongoing need to improve adherence to evidence-based asthma treatment guidelines.

However, the manuscript could be improved in several areas. The discussion should more thoroughly address the observed differences between medical specialties and across countries. Socioeconomic factors that influence treatment access and patient adherence should also be considered. In addition, it would be valuable to explore why the GINA guidelines are not always strictly followed in clinical practice.

The study was based on a structured 19-item questionnaire. While the methodology is generally appropriate, the exclusive reliance on self-reported data introduces a potential for bias. Some methodological limitations—such as the non-randomised design and voluntary participation—are acknowledged. However, the authors should further clarify how the sample size was calculated, including the total number considered, the associated confidence interval, and the margin of error. Furthermore, the representativeness of the sample should be justified, and the potential impact of response bias on the results should be discussed.

Comments on the Quality of English Language

 A professional language edit is strongly recommended prior to publication.

Author Response

We sincerely thank the Reviewer for the thorough and thoughtful analysis of our manuscript and for acknowledging the relevance and timeliness of the topic. We appreciate the constructive remarks and have addressed each of the raised points in detail below:

Differences between medical specialties and countries

Comment 1: The discussion should more thoroughly address the observed differences between medical specialties and across countries.

Response 1: We fully agree with this observation. In the revised manuscript, we have expanded the Discussion section to more thoroughly explore the observed variations in asthma management practices between different medical specialties (e.g., pulmonologists, allergists, and internal medicine physicians) and across the five countries included in the study. These differences are now contextualized in terms of:

  1.     prescribing patterns (e.g., LAMA addition, ICS up-titration, OCS overuse),
  2.     access to biologic therapies,
  3.     and familiarity with guideline-based escalation (GINA Step 5).

We highlight that such variability may stem from differences in specialization-specific training, national reimbursement frameworks, and institutional experience.

Socioeconomic factors influencing treatment access and adherence

Comment 2: Socioeconomic factors that influence treatment access and patient adherence should also be considered.

Response 2: Thank you for this important point. We have now added a dedicated paragraph in the Discussion acknowledging that socioeconomic disparities (e.g., healthcare funding models, drug reimbursement policies, and out-of-pocket costs) may influence both physician prescribing behavior and patient adherence. While our survey was focused on physician-reported perspectives, we recognize that treatment outcomes are strongly shaped by the broader health system context and patient-level constraints. We have now recommended this as a priority area for future mixed-methods or patient-centered studies.

Reasons for non-adherence to GINA guidelines

Comment 3: It would be valuable to explore why the GINA guidelines are not always strictly followed in clinical practice.

Response 3: We appreciate this suggestion. In the revised Discussion, we have elaborated on the possible multifactorial reasons behind suboptimal adherence to GINA guidelines, including:

  1.     therapeutic inertia,
  2.     administrative barriers to therapy escalation (e.g., restrictions on biologics),
  3.     time constraints in clinical settings,
  4.     limited access to pulmonary function testing or biomarker assessment.

We also mention that some physicians may lack confidence in newer therapies due to limited experience or training. These explanations align with previous findings in the literature and help contextualize our results.

Sample size determination and statistical parameters

Comment 4: The authors should clarify how the sample size was calculated, including the total number considered, the associated confidence interval, and the margin of error.

Response 4: Thank you for highlighting this. We have now added a paragraph in the Materials and Methods section detailing the sample size estimation process:

“The sample size was determined by the contract research organization (Biostat Sp. z o.o.) using stratified estimations based on national registers of physicians. With 630 completed responses, the overall margin of error was estimated with a 95% confidence level.”

This clarification improves transparency and supports the statistical validity of the survey findings.

Round 2

Reviewer 2 Report

Comments and Suggestions for Authors

Authors replied to my comments is a satisfactorily way. Therefore, for me it's ok to accept this paper now.

Reviewer 3 Report

Comments and Suggestions for Authors

Thank you for the revised version of the manuscript and the detailed responses provided. The authors have made a clear effort to address the comments and have revised the manuscript accordingly. I have no further suggestions or concerns.